# Diagnosis, Prognosis, and Treatment of Canine Hemangiosarcoma: A Review Based on a Consensus Organized by the Brazilian Association of Veterinary Oncology, ABROVET

**DOI:** 10.3390/cancers15072025

**Published:** 2023-03-29

**Authors:** Andrigo Barboza De Nardi, Cristina de Oliveira Massoco Salles Gomes, Carlos Eduardo Fonseca-Alves, Felipe Noleto de Paiva, Laís Calazans Menescal Linhares, Gabriel João Unger Carra, Rodrigo dos Santos Horta, Felipe Augusto Ruiz Sueiro, Paulo Cesar Jark, Adriana Tomoko Nishiya, Carmen Helena de Carvalho Vasconcellos, Rodrigo Ubukata, Karen Batschinski, Renata Afonso Sobral, Simone Crestoni Fernandes, Luiz Roberto Biondi, Ricardo De Francisco Strefezzi, Julia Maria Matera, Marcelo Monte Mor Rangel, Denner Santos dos Anjos, Carlos Henrique Maciel Brunner, Renee Laufer-Amorim, Karine Germano Cadrobbi, Juliana Vieira Cirillo, Mauro Caldas Martins, Nazilton de Paula Reis Filho, Diego Fernando Silva Lessa, Roberta Portela, Carolina Scarpa Carneiro, Sílvia Regina Ricci Lucas, Heidge Fukumasu, Marcus Antônio Rossi Feliciano, Juliany Gomes Quitzan, Maria Lucia Zaidan Dagli

**Affiliations:** 1Department of Veterinary Clinic and Surgery, Universidade Estadual Paulista (UNESP), Jaboticabal 14884-900, Brazil; 2Department of Pathology, School of Veterinary Medicine and Animal Science, Universidade de São Paulo (USP), São Paulo 05508-900, Brazil; 3Institute of Health Sciences, Universidade Paulista (UNIP), Bauru 17048-290, Brazil; 4Department of Veterinary Surgery and Animal Reproduction, Universidade Estadual Paulista (UNESP), Botucatu 18618-681, Brazil; 5Department of Veterinary Medicine and Surgery, Veterinary School, Universidade Federal de Minas Gerais, Belo Horizonte 31270-901, Brazil; 6Histopathological Diagnosis Department, VETPAT—Animal Pathology & Molecular Biology, Campinas 13073-022, Brazil; 7Onccarevet e Onconnectionvet Clinic, Ribeirao Preto 14026-587, Brazil; 8Naya Specialties, Campo Belo 04608-003, Brazil; 9Surgical and Clinical Oncology, Botafogo Veterinary Hospital, Rio de Janeiro 22281-180, Brazil; 10Clinical and Surgical Oncology, E+ Veterinary Specialties, São Paulo 04078-012, Brazil; 11Clinical, Surgical and Palliative Care Oncology, Onco Cane Veterinary, São Paulo 04084-002, Brazil; 12SEOVET—Specialized Service in Veterinary Oncology, Clinical and Surgical Oncology, São Paulo 05016-000, Brazil; 13Small Animal Internal Medicine Department, School of Veterinary Medicine, Universidade Metropolitana de Santos (UNIMES), Santos 11045-002, Brazil; 14Department of Veterinary Medicine, Faculty of Animal Science and Food Engineering Universidade de São Paulo (USP), Pirassununga 13635-900, Brazil; 15Department of Surgery, School of Veterinary Medicine and Animal Science, Universidade de São Paulo (USP), São Paulo 05508-270, Brazil; 16Clinical and Surgical Oncology, Vet Cancer Animal Oncology and Pathology, São Paulo 04523-013, Brazil; 17Eletro-Onkovet Service, Franca 14406-005, Brazil; 18Department of Veterinary Clinic, School of Veterinary Science and Animal Health, Universidade Estadual Paulista (UNESP), Botucatu 18618-681, Brazil; 19CRV Image—Veterinary Reference Center, Rio de Janeiro 22631-100, Brazil; 20NeoPet Vet Oncology, Londrina 86065-435, Brazil; 21Lessa Cardiology Vet, Ribeirao Preto 14020-180, Brazil; 22MedVet Chicago, Chicago, IL 60618, USA; 23City Hall of São Paulo, Municipal of Health of the State of São Paulo, São Paulo 01223-011, Brazil; 24Department of Internal Medicine, School of Veterinary Medicine and Animal Science, Universidade de São Paulo (USP), São Paulo 05508-000, Brazil

**Keywords:** endothelial tumors, angiosarcoma, guidelines, dog, visceral, actinic, cutaneous

## Abstract

**Simple Summary:**

Hemangiosarcoma is a mesenchymal neoplasm that originates in the endothelial cells of blood vessels. According to the location of origin, they can be classified as non-visceral and visceral types. Hemangiosarcoma can be very aggressive and metastasize to distant organs. The aim of this critical review is to present data on the epidemiology, etiology, diagnosis, staging, therapeutic modalities, and prognosis of canine hemangiosarcoma based on a consensus meeting organized by the Brazilian Association of Veterinary Oncology (ABROVET) in September 2022. Recent information from the literature, as well as new results from consensus participants, are presented and discussed.

**Abstract:**

Hemangiosarcoma is a mesenchymal neoplasm originating in the endothelial cells of blood vessels; they can be classified as non-visceral and visceral types. Non-visceral hemangiosarcomas can affect the skin, subcutaneous tissues, and muscle tissues; visceral hemangiosarcomas can affect the spleen, liver, heart, lungs, kidneys, oral cavity, bones, bladder, uterus, tongue, and retroperitoneum. Among domestic species, dogs are most affected by cutaneous HSA. Cutaneous HSA represents approximately 14% of all HSA diagnosed in this species and less than 5% of dermal tumors, according to North American studies. However, Brazilian epidemiological data demonstrate a higher prevalence, which may represent 27 to 80% of all canine HSAs and 13.9% of all skin neoplasms diagnosed in this species. Cutaneous HSA most commonly affects middle-aged to elderly dogs (between 8 and 15 years old), with no gender predisposition for either the actinic or non-actinic forms. The higher prevalence of cutaneous HSA in some canine breeds is related to lower protection from solar radiation, as low skin pigmentation and hair coverage lead to greater sun exposure. Actinic changes, such as solar dermatosis, are frequent in these patients, confirming the influence of solar radiation on the development of this neoplasm. There are multiple clinical manifestations of hemangiosarcoma in canines. The diagnostic approach and staging classification of cutaneous HSAs are similar between the different subtypes. The definitive diagnosis is obtained through histopathological analysis of incisional or excisional biopsies. Cytology can be used as a presurgical screening test; however, it has little diagnostic utility in cases of HSA because there is a high risk of blood contamination and sample hemodilution. Surgery is generally the treatment of choice for dogs with localized non-visceral HSA without evidence of metastatic disease. Recently, electrochemotherapy (ECT) has emerged as an alternative therapy for the local ablative treatment of different neoplastic types; the use of radiotherapy for the treatment of dogs with cutaneous HSA is uncommon. There is greater consensus in the literature regarding the indications for adjuvant chemotherapy in subcutaneous and muscular HSA; doxorubicin is the most frequently used antineoplastic agent for subcutaneous and muscular subtypes and can be administered alone or in combination with other drugs. Other therapies include antiangiogenic therapy, photodynamic therapy, the association of chemotherapy with the metronomic dose, targeted therapies, and natural products. The benefits of these therapies are presented and discussed. In general, the prognosis of splenic and cardiac HSA is unfavorable. As a challenging neoplasm, studies of new protocols and treatment modalities are necessary to control this aggressive disease.

## 1. Introduction

The Brazilian Association of Veterinary Oncology (ABROVET) organized a Consensus on Hemangiosarcoma on 17 and 18 September 2022. The meeting joined 20 speakers, known veterinary oncologists from Brazil and the USA (Roberta Portela), and 351 veterinarians and veterinary students interested in veterinary oncology who attended the online meeting. This review is based on the presentations and discussions raised on the consensus.

## 2. Canine Hemangiosarcoma and Its Different Forms

Hemangiosarcoma (HSA) is a mesenchymal neoplasm originating from the endothelial cells of the blood vessels. It is also called a malignant hemangioendothelioma or angiosarcoma, or Visceral Vascular Tumor, which includes hemangiomas and hemangiosarcomas. The benign counterpart of hemangiosarcoma is hemangioma [1].

Hemangiosarcomas are categorized into two types according to their location or origin: non-visceral hemangiosarcomas and visceral hemangiosarcomas. Non-visceral hemangiosarcomas can affect the skin, subcutaneous tissues, and muscle tissues (Figure 1). In contrast, visceral hemangiosarcomas can affect the spleen, liver, heart, lungs, kidneys, oral cavity, bones, bladder, uterus, tongue, and retroperitoneum [1].

## 3. Cutaneous Hemangiosarcoma

Canine non-visceral HSA develops more frequently in cutaneous tissues (the dermis and hypodermis). Cutaneous HSA can arise primarily in the dermis, with or without invasion into subcutaneous and muscle tissues, or in the hypodermis (subcutaneous), with or without invasion into muscle tissues. Primary muscular HSA is considered less common [1,2]. Cutaneous HSA, primarily dermal, is subdivided into actinic and non-actinic HSA.

Non-visceral HSA can be divided into four main subtypes based on biological differences: actinic cutaneous, non-actinic cutaneous, subcutaneous, and muscular (Figure 2).

## 4. Etiology and Epidemiology

### 4.1. Actinic and Non-Actinic Cutaneous HSA

Among domestic species, dogs are most affected by cutaneous HSA. Cutaneous HSA represents approximately 14% of all HSA diagnosed in this species and less than 5% of dermal tumors, according to North American studies [3,4,5]. However, Brazilian epidemiological data demonstrate a higher prevalence, which may represent 27 to 80% of all canine HSAs [6,7] and 13.9% of all skin neoplasms diagnosed in this species [8]. This difference is mainly associated with the substantial incidence of solar radiation in countries with tropical climates [9]. According to data from the National Institute for Space Research [10], Brazil has a high level of ultraviolet radiation, exceeding a scale of 6.0 during most of the year, a factor directly related to the etiology of cutaneous actinic HSA. It is important to note, however, that these epidemiological studies did not differentiate between the actinic and non-actinic subtypes.

Cutaneous HSA most commonly affects middle-aged to elderly dogs (between 8 and 15 years old), with no gender predisposition for either the actinic or non-actinic forms [2,11,12]. Dogs of breeds with lightly pigmented and glabrous skin are at higher risk of developing cutaneous HSA, mainly in its actinic form. The most reported breeds include Pitbull, Whippet, Greyhound, Boxer, Beagle, and Dalmatian [2,3,11,13].

The higher prevalence of cutaneous HSA in these animals is related to lower protection from solar radiation, as low skin pigmentation and hair coverage lead to greater sun exposure. Actinic changes, such as solar dermatosis, are frequent in these patients, confirming the influence of solar radiation on the development of this neoplasm [11,13,14]. Such changes occur progressively, and often, affected dogs initially develop hemangiomas that can progress to HSA upon undergoing malignant transformation [15].

Acute exposure to ultraviolet B (UVB) radiation causes skin inflammation and oxidative stress, and long-term exposure to UVB radiation can lead to carcinogenesis. Evidence has demonstrated that reactive oxygen species (ROS) constitute the link between chronic inflammation and neoplasia. Initial experiments on the role of ROS in tumor initiation indicated that oxidative stress directly damages DNA, promoting mutations that favor oncogenic transformation [16,17].

Non-actinic cutaneous HSA, in turn, occurs more frequently in dogs of non-predisposed breeds that have pigmented skin and thick coats, as well as in those without a history of chronic sun exposure. Therefore, in the histopathological analysis of these tumors, actinic alterations are not identified [11]. Although the origin of non-actinic HSA remains unknown, solar radiation is unlikely to play a role in its development, and the etiopathogenesis is likely similar to that of visceral HAS [18]. Further studies are required to confirm this hypothesis.

Several studies have investigated the etiopathogenesis of this neoplasm by analyzing genomic profiles to identify mutations in specific genes. Similar to visceral HSA, the most frequently mutated gene in canine cutaneous HSA is *TP53* [19]. However, it is important to highlight that *TP53* mutations are important and frequently found in several cancer subtypes. García-Iglesias et al. [20] found a high expression of the mutated TP53 gene associated with high Ki-67 proliferative activity in 25 dogs with cutaneous HSA, suggesting the association of this gene mutation with neoplastic development. Both studies included patients with dermal and hypodermic HSAs. However, further studies are necessary to elucidate the TP53 role in canine HAS development.

Dysregulation of the PI3K/AKT/mTOR pathway is also very important in canine tumors [21]. Kim et al. [18] evaluated the comparative genomic alteration in human and canine angiosarcomas and found *TP53, PIK3CA, PIK3R1,* and *NRAS* in a specific set of canine hemangiosarcomas suggesting these genes dysregulation could be important for tumor development [18].

Mutations in PTEN genes and overexpression of angiogenesis-related growth factors, such as vascular endothelial growth factor (VEGF) and basic fibroblast growth factor (bFGF), have also been identified in cutaneous and subcutaneous HSA [2,19,21,22], suggesting participation in the etiopathogenesis of the disease.

#### Subcutaneous and Muscular HSA

Compared with dermal-confined cutaneous HSA, few studies have exclusively evaluated subcutaneous and muscular HSAs. These subtypes account for approximately 6–47% of canine HSAs, affecting middle-aged to elderly dogs (mean age 9 years), with no gender predisposition [2,11,13,23,24,25,26].

Due to the scarcity of data in the current literature, it is not yet possible to identify a significant breed predisposition. However, studies have reported a higher incidence of subcutaneous and muscular HSA in golden retrievers, labrador retrievers, and mixed-breed dogs [25,26,27]. Furthermore, unlike actinic cutaneous HSA and similar to non-actinic cutaneous HSA, the development of these subtypes does not seem to correlate with exposure to solar radiation, and their etiopathogenesis may be related to the same factors involved in the development of visceral HSA [13,18,28].

## 5. Clinical Manifestation and Biological Behavior

### Actinic and Non-Actinic Cutaneous HSA

The clinical symptoms in dogs with non-visceral HSA vary according to the primary location of the neoplasm and local infiltration. Actinic cutaneous HSA appears more frequently in the ventroabdominal (Figure 3), preputial, and pelvic limb regions and in glabrous, short-haired dogs with a history of chronic sun exposure [2,3,11].

Generally, dermal cutaneous HSA presents as solitary or multiple superficial nodules or papules with a reddish-to-blackish color. Clinical manifestations are usually local and limited to slight intermittent bleeding in the tumor region [1,3]. Studies have identified a higher incidence (28–35%) of the development of multiple lesions of actinic cutaneous HSA in dogs of predisposed breeds [3,11].

Despite its exact histological origin, the biological behavior of actinic cutaneous HSA differs from that of non-actinic HSA. In dogs that develop the actinic subtype, the disease tends to manifest in a less aggressive course with a lower probability of metastatic development and longer survival. In contrast, non-actinic cutaneous HSA is associated with a higher metastatic rate and lower survival rate, with a higher incidence in non-predisposed breeds [2,11].

## 6. Subcutaneous and Muscular HSA

Subcutaneous and muscular HSAs present as more significant, adherent, or mobile than actinic or cutaneous, with a firm-to-soft consistency, and may be associated with ulcerations [1,3,13,25].

Unlike the actinic cutaneous subtype, subcutaneous and muscular HSAs do not have an anatomical predilection for their development and may appear in different locations, including the limbs, flank, trunk, scapula, and cervical regions [25,26,27]. Affected animals may develop local pain, lameness, or functional impairment of other structures depending on the anatomical location of the tumor [3,15,26].

Local hemorrhages in subcutaneous tissue and muscles can occur in dogs with advanced HSA [1,26]. As in visceral HSA, the blood vessels associated with the tumor are deformed and tortuous, leading to blood deprivation for tumor cells and resulting in death. This event causes small foci of vascular rupture, culminating in blood leakage into the subcutaneous or intramuscular space [18].

Subcutaneous and muscular HSAs are naturally more aggressive than the dermal subtype since these subtypes have infiltrative growth involving deeper tissues and greater metastatic capacity [3,26,29]. However, no studies have compared the behavior of non-actinic cutaneous HSA with those of subcutaneous and muscular HSA.

## 7. Diagnosis and Staging

The diagnostic approach and staging classification of cutaneous HSAs are similar between the different subtypes. The definitive diagnosis of these tumors is obtained through histopathological analysis of incisional or excisional biopsies. Cytology can be used as a presurgical screening test; however, it has little diagnostic utility in cases of HSA because there is a high risk of blood contamination and sample hemodilution. For this reason, in cytological analysis, cutaneous HSA can be easily confused with hemangioma, a benign neoplasm considered one of the main differential diagnoses [1,30]. The presence of mast cell aggregates in cytological samples can mimic a mast cell tumor and be a confounding factor for pathologists [29].

For cutaneous HSA, the association between the classic macroscopic presentation of the lesions and a cytological report containing malignant mesenchymal cells represents a strong indication of HSA. In contrast, subcutaneous and muscular HSAs do not usually manifest as reddish to violaceous tumors and thus are more easily confused with other tumor types of subcutaneous origin or involvement.

Despite the differences in biological behavior between HSA subtypes, the cell morphology, histological organization, and pathobiological characteristics are similar, usually following an infiltrative growth pattern in the surrounding tissues and sometimes causing hemorrhage and tissue necrosis [18]. Histologically, actinic changes have been identified in cutaneous actinic HSA lesions, including solar elastosis, dermatitis, superficial dermal fibrosis, and epidermal dysplasia [11].

HSA presents as a heterogeneous neoplasm and may contain areas with well-differentiated endothelial cells, forming well-defined vascular channels, as well as solid areas composed of highly anaplastic endothelial cells. This latter scenario can make it difficult to differentiate it from other sarcomas, requiring immunohistochemical diagnosis [30,31].

To date, there is no standardization of the histopathological criteria for cutaneous, subcutaneous, and muscular HAS [2,24,32,33]. Initially, Hammer et al. (1991) [32] applied the grading system for soft tissue sarcomas proposed by Russel et al. (1977) [34] to dogs with HSA but found no association between grade and prognosis. Subsequently, Olgivie et al. (1996) [24] proposed a histological grading system in dogs with visceral and non-visceral HSA, classifying them as low-, intermediate-, or high-grade, and dogs with low-grade tumors had a better prognosis. Wendelburg et al. (2015) [33] used the same grading system and found no association between tumor grading and prognosis.

In a recent study, Nóbrega et al. (2019) [2] evaluated 60 dogs with non-visceral HSA (41 with cutaneous, 14 with subcutaneous, and 4 with muscular involvement), classifying the samples into three different scores according to the degree of tumor differentiation, quantification of vascular channels, and variations in nucleus morphology (score 1, well-differentiated; score 2, moderately differentiated; and score 3, poorly differentiated). However, there was no association between the degree of differentiation and the survival of these patients.

There is still no consensus on a cut-off mitotic index value for the number of mitoses, which would make it possible to classify the proliferation of HSA and determine its prognostic value. Shiu et al. (2011) [26] found no correlation between the mitotic index and survival in 71 dogs with subcutaneous and intramuscular HSA. Nóbrega et al. (2019) [2] evaluated the mitotic index in dogs with cutaneous HSA by classifying tumors into four groups (1–5 mitoses, 5–10 mitoses, 11–20 mitoses, and >20 mitoses), but they did not define its correlation with survival time.

New studies are needed to define and standardize different criteria for the histopathological classification of HSA, mainly regarding the histopathological grade and mitotic index. These tools are of great value for characterizing the biology of tumors and obtaining prognostic information. It is also of great value that key points are determined in the histopathological evaluation to seek greater standardization between reports, thus allowing the conduction of new clinical studies with more homogeneous criteria. In this context, Szivek et al. (2011) [11] emphasized the importance of describing the degree of neoplastic invasion and the presence of actinic lesions in the histopathological report since the identification of subcutaneous invasion was associated with a risk twice as high as that associated with metastatic dissemination, and the absence of lesions actinic cells was associated with a higher metastatic rate and shorter survival.

Owing to the cavitary nature of malignant vascular neoplasms, it can be difficult to differentiate between well-differentiated HSA and hemangioma [13] and between poorly differentiated HSA and other types of sarcomas [31]. Immunohistochemical evaluation is indicated in these cases. Several studies have identified immunohistochemical markers for the diagnosis of HSA, including the Willebrand factor (FVIII), VEGF, bFGF, and claudin-5 [22,31,35,36]. In addition, García-Iglesias et al. (2020) [20] demonstrated that CD117 immunoexpression (KIT) and Ki-67 index could be useful in differentiating between cutaneous HSA and hemangioma.

Complete staging of patients with non-visceral HSA is fundamental, allowing knowledge of the local and systemic extent of the disease and the identification of possible comorbidities, which is essential for therapeutic decision-making. Laboratory and imaging tests are indicated [5,15,37].

Laboratory alterations in dogs affected by cutaneous HSA occur infrequently and are less specific than those found in patients with visceral HSA. Szivek et al. (2011) [11] found that 2 of 82 dogs with cutaneous HSA had anemia, and 1 had anemia associated with thrombocytopenia. Shiu et al. (2011) [26] evaluated 70 dogs with subcutaneous and intramuscular HSAs and identified anemia and thrombocytopenia in 28% and 25% of patients, respectively. In that study, anemic animals had significantly lower survival times than non-anemic animals. Anemia in these patients may be due to chronic inflammation, hemolysis, hemorrhage, or bone marrow dysfunction [1].

Shiu et al. (2011) [26] found that only 26% of dogs had neutrophilia. Neutrophilic leukocytosis may occur in cases of cutaneous HSA resulting from the presence of inflammation and tumor necrosis and is more commonly observed in dogs with larger and infiltrative tumors.

Although less frequent, thrombocytopenia may be secondary to acute hemorrhage, intratumoral destruction, and coagulation consumption. However, disseminated intravascular coagulation (DIC) is uncommon in patients with cutaneous HSA [1]. Alterations in biochemical dosage have rarely been reported in the literature [11,26,38]. However, these tests should be routinely performed to identify possible comorbidities.

Abdominal ultrasonography and chest radiography in three projections (ventrodorsal, left, and right lateral) are the imaging modalities most commonly used to screen for cancer staging in veterinary patients to identify possible distant metastases [5,39].

Cutaneous HSA can occur concomitantly with visceral HSA and may also represent metastases from systemic HSA [15,40]. Therefore, complete and more extensive staging is important for subcutaneous, muscular, and non-actinic cutaneous HAS [11,26].

Computed tomography may be indicated for surgical planning when tumors are large and infiltrative and for detecting metastases, including the search for pulmonary nodules in patients with chest X-rays without apparent lesions [1,39,41,42]. Dogs diagnosed with hemangiosarcoma were more prone to the appearance of signs compatible with pulmonary metastasis in the first evaluation by computed tomography, justifying the importance and use of this technique in oncological staging [43].

Staging based only on the correlation between ultrasound and radiographic examinations is not capable of accurately detecting muscle metastases, especially in patients with nonspecific clinical signs who present inconclusive neurological or orthopedic examinations [39]. The lack of specific signs associated with metastases supports the use of full-body contrast-enhanced computed tomography. In a study of 61 dogs with hemangiosarcoma, all patients with muscle metastasis had advanced stages of the disease [44].

Whole-body tomographic study guides the collection of material from the lesions (by cytology or biopsy), providing more information for clinical assistance in managing and monitoring patients with hemangiosarcomas [39,44].

Table 1 describes the clinical staging system used for cutaneous HSA. The current system can classify non-actinic cutaneous HSA at an early stage, which should be interpreted with caution since these have a more aggressive biological behavior than their actinic form.

## 8. Treatment

### Local Therapies

Surgery is generally the treatment of choice for dogs with localized non-visceral HSA without evidence of metastatic disease. Therefore, cutaneous, subcutaneous, and muscular subtypes are addressed in the same subtopic [1].

For dogs with cutaneous HSA (stage I), surgical resection with lateral margins of 1–2 cm and deep margins in the fascial plane is recommended. Patients can be cured when complete lesions are removed, particularly in cases of cutaneous HSA with actinic components [3,5,11]. Recent studies have evaluated the application of proportional or “mirrored” lateral margins in mast cell tumors <2 cm, obtaining satisfactory results concerning local control [45,46]. This approach can also be considered for cutaneous HSA, particularly in cases where multiple small lesions are present. However, further studies are required to better define this criterion for cutaneous HSA.

When the tumors are more extensive and infiltrative, which occurs mainly in cases involving the subcutaneous and muscular tissues, more extensive surgeries are necessary and recommend lateral surgical margins of at least 3 cm, similar to those adopted in sarcomas of soft tissues [1,5].

Widely used in human medicine, transoperative histological evaluation by freezing has gained popularity in veterinary medicine and can be used to confirm the diagnosis of cutaneous HSA, as well as to identify neoplastic cells in the lateral and deep surgical margins, helping to determine the extent of the procedure [47,48].

Given the scarcity of data in the literature about the rate of lymphatic metastases by HSA, there is limited information about the real benefit of performing lymphadenectomy at the time of surgery. Therefore, lymphadenectomy is recommended only in specific cases in which at least one of the following criteria are present: (1) neoplastic infiltration in a lymph node, detected by previous FNAB; (2) changes in palpation examination (volume, shape, consistency, and/or adherence); and (3) morphological alterations identified through imaging tests (abdominal ultrasound or computed tomography).

Obtaining wide margins becomes difficult when developing multiple large tumors and/or in complex anatomical sites, which may require more radical surgical procedures [1,3]. Alternatively, for these cases, less extensive surgical approaches can be associated with local and systemic therapies such as electrochemotherapy, radiotherapy, or adjuvant chemotherapy [27,49,50].

The use of radiotherapy for the treatment of dogs with cutaneous HSA is uncommon, and only a few studies have evaluated its clinical benefits. Studies published so far suggest the use of palliative radiotherapy in cases of unresectable tumors, with the aim to reduce pain, possible bleeding, and tumor progression; post-surgical radiotherapy is also used for the treatment of compromised margins [1,26,28,50].

Hillers et al. (2007) [50] evaluated the effect of palliative radiotherapy (6–24 Gy) in 20 dogs with unresectable subcutaneous and intramuscular HSA with or without surgery and chemotherapy and found an overall response rate of 70% (10 PR and 4 CR); however, there was no increase in overall survival time.

The efficacy of radiotherapy in the treatment of microscopic disease after incomplete surgical resection in dogs with subcutaneous and intramuscular HSA remains controversial. Bulakowski et al. (2008) [25] administered post-surgical radiotherapy to five dogs and found a short duration of response, and local recurrence occurred in three of these five dogs. Shiu et al. (2011) [26] found that the three dogs treated with radiotherapy for the same purpose had no local recurrence. Larger prospective studies are needed to determine the effectiveness of radiotherapy in the local control of subcutaneous and intramuscular HSA.

Recently, electrochemotherapy (ECT) has emerged as an alternative therapy for the local ablative treatment of different neoplastic types [51,52,53], and its application has already been standardized in Europe for cutaneous and subcutaneous tumors of different histological types in humans, canines, and felines [51]. ECT consists of a chemotherapeutic drug, generally of a hydrophilic nature, whose penetration through the plasma membrane is limited, used with electrical pulses; the electrical pulses provide a temporary and reversible increase in the permeability of the target cell membrane, maximizing the intracellular concentration of the previously applied chemotherapy, and consequently, its cytotoxic effect [54,55]. The main antineoplastic drugs used are bleomycin and cisplatin, which can be administered intravenously, intratumorally, or intratumorally.

Bleomycin (BLM) is the most commonly used chemotherapeutic in ECT protocols because the internalization of its molecules is enhanced by approximately 300 to 700 times in the intracellular environment [56]. BLM has the possibility of intravenous administration, allowing its homogeneous distribution in the area to be treated and providing unique selectivity. This is attributed to its cell death mechanism, which provides selective destruction of cells that are in high mitotic activity (tumor cells) and allows the treatment of margins, preserving healthy tissue and eliminating possible microscopic neoplastic foci present there [51,57].

In addition to its cytotoxic antitumor effect, ECT exerts a hemostatic action on the tumor vasculature through local vasoconstriction. This effect promotes a reduction in intraoperative bleeding when used as an adjunct to surgery and as a monotherapy in multiple and hemorrhagic lesions [58].

Campana et al. (2019) [59] evaluated 20 human patients with 51 target lesions who underwent ECT in cutaneous HSA in advanced stages, in which surgery became unfeasible due to local spread. CR was observed in 61%, PR in 22%, DE in 18%, and DP in 2% of injuries. CR was observed in 40% (8/20), PR in 40% (8/20), DE in 15% (3/20), and DP in 5% (1/20) of patients. Patients who achieved CR had considerable local control with an average progression-free time of 10.9 months. Regarding the antivascular effect, a reduction in bleeding was observed in 93% of the patients (13/14) who had an ulcerated tumor, providing significant palliative benefits and promoting quality of life. However, 35% of patients (n = 7) had tumor recurrence after 3.4 months. Palliative ECT has been used to improve quality of life and life expectancy.

Several studies have already evaluated the effectiveness of ECT in different types of canine sarcomas, mostly soft tissue sarcomas (STMs), with promising results [60,61,62,63,64]. However, owing to the small population of dogs with cutaneous HSA in these studies, it is still impossible to comprehensively determine the effectiveness of ECT in these tumors.

In most studies performed with ECT in cutaneous hemangiosarcomas, the therapy is used almost exclusively as an adjuvant to surgical treatment and has a limited response to its use alone [63]. However, despite the lack of a large sample group, the clinical benefit of ECT therapy seems promising in terms of local control and decreased recurrence when used together with surgery [63].

In a recent study of 30 dogs with soft tissue sarcoma, only 2 had a diagnosis of cutaneous HSA with a DFI of 1053 and 366 days, since the second dog died due to recurrence and metastasis at that time [63].

Spugnini et al. [64] observed 22 dogs with soft tissue sarcoma, and 3 were diagnosed with HSA in limited anatomical regions regarding the practical surgical approach (limbs, pelvis, and thorax). All patients had local control but a short survival time (30, 60, and 150 days) with recurrence, metastasis, and death from splenic HSA.

The distribution of intratumoral BLM is affected by the intrinsic vascularization of the tumor. Therefore, the results of electroporation associated with chemotherapy combinations vary. In a study of human patients, it was observed that patients diagnosed with cutaneous angiosarcoma had the highest concentrations of BLM (819 mg/g); however, the tumor presented tumor progression and had no response to ECT [65,66], showing a difference in perfusion, vascularization, and response in different tumor types.

As observed in human patients, cutaneous toxicity is also observed when using ECT alone, with varying degrees of cutaneous ulceration [59] (personal communication) (Figure 4 and Figure 5).

Based on available evidence, ECT may be considered in specific cases of cutaneous HSA: (1) as monotherapy, in multiple and disseminated lesions, common in actinic cutaneous HSA; (2) intraoperatively, in cases where it is challenging to obtain margins due to extension, number of lesions, or tumor location; (3) alternatively, in the treatment of compromised or narrow margins, in cases in which the surgical enlargement of margins is unfeasible due to anatomical limitations; and (4) as a factor for reducing tumor extension prior to surgery in places where primary excision with adequate margins would cause significant anatomical changes in the region. Larger studies are required to better define its efficacy.

Cryosurgery has also been used to treat cutaneous HSA. Despite its everyday use in clinical practice, only a limited number of previously published studies have evaluated the disease-free interval and overall survival of patients with cutaneous HSA. Most relevant studies have evaluated the effect of cryosurgery on cutaneous tumors and were not specific to HAS [67,68]. This technique appears promising based on previous studies; however, future clinical studies are needed to confirm its effectiveness.

## 9. Systemic Therapies

### 9.1. Actinic and Non-Actinic Cutaneous HSA

Unlike in cases of visceral HSA, surgical treatment is considered curative for dogs with actinic cutaneous HSA, confined to the dermis, and without metastatic evidence (stage I), with no indication of adjuvant chemotherapy for these patients.

However, for cases of non-actinic HSA, there is still no consensus regarding the indications for adjuvant chemotherapy. Given the more aggressive and metastatic behavior of non-actinic HSA, studies must be conducted to determine the clinical benefits of systemic therapy in these patients [11].

### 9.2. HSA Subcutaneous and Muscular

There is greater consensus in the literature regarding the indications for adjuvant chemotherapy in subcutaneous and muscular HSA [5]. This recommendation is based on the more aggressive biological behavior inherent in these subtypes, both because of their infiltrative nature, making surgical resection and local control more complex, and because of their significant metastatic potential, resulting in shorter survival times and, consequently, worse prognoses [3,11,26,69].

Similar to visceral HSA, doxorubicin is the most frequently used antineoplastic agent for subcutaneous and muscular subtypes and can be administered alone or in combination with other drugs [23,24,25,32,69]. Retrospective studies that evaluated the use of chemotherapy in animals with subcutaneous and intramuscular HSAs found median survival rates between 210 and 1189 days (Table 2).

Hammer et al. (1991) [32] treated four dogs with subcutaneous HSAs with incomplete surgical resection with the VAC protocol, resulting in a median survival of 436 days. Sorenmo et al. (1993) [23] evaluated the AC protocol in 5 dogs with subcutaneous HSAs (complete resection in 2/5 dogs), which achieved a median survival of 240 days. Bukowski et al. (2008) [25] evaluated chemotherapy in only dogs with subcutaneous and intramuscular HSA (n = 21) after adequate local surgical control, using doxorubicin with or without cyclophosphamide; 17 patients with subcutaneous HSA had significantly higher median survival than those with intramuscular HSA (1189 vs. 272 days, respectively).

Shiu et al. (2011) [26] studied 55 dogs with subcutaneous HSAs; 17 received adjuvant chemotherapy (protocols with doxorubicin combined with other drugs) after complete surgical resection and had a median survival of 246 days. The type of HSA and adjuvant treatment did not influence the clinical outcomes of patients. However, dogs that underwent total surgical excision of the lesions had a significantly higher median survival and progression-free time than those with incomplete resection.

Prospective studies in dogs with stage II and III HSA are needed to better determine the therapeutic benefit of adjuvant chemotherapy. However, given the more advanced staging and more aggressive course, the authors of this consensus recommend using adjuvant chemotherapy in these cases, with doxorubicin as the primary drug.

Aggressive and infiltrative subcutaneous and intramuscular HSAs can be unresectable, making surgical treatment unfeasible in affected animals. For these cases, chemotherapy can be instituted in two different situations: (1) neoadjuvant to surgery (pre-surgical) to reduce tumor dimensions to make surgical excision possible, either with curative or palliative intent, or (2) purely palliatively, without association with surgery, to offer comfort to patients with advanced and unresectable tumors.

In this context, Wiley et al. (2010) [27] evaluated the benefits of chemotherapy with or without doxorubicin in 18 canine patients with unresectable subcutaneous HSA. The dogs received surgical treatment after chemotherapy (neoadjuvant scenario, n = 5) or chemotherapy alone (palliative scenario, n = 13). Doxorubicin was administered in all protocols as a single drug (12/18) or in combination with other chemotherapeutics (6/18) every 2 or 3 weeks. The overall response rate ranged from 38 to 44%, and tumor excision was achieved with wide margins in 80% of the patients who underwent surgery after chemotherapy. Although the response was short-lived, neoadjuvant chemotherapy provided clinical benefits for these patients, and surgical excision was associated with a longer progression-free time.

### 9.3. Other Therapies

Considering the vascular nature of HSAs, antiangiogenic therapy is a potential therapeutic target for these tumors in dogs, whether in their visceral or cutaneous form [12,70]. Its use has been widely investigated, mainly in splenic HSA, and studies are needed to assess its clinical benefit in canine cutaneous HSA [12,70,71].

In recent years, the search for molecular therapeutic targets in visceral and non-visceral HSAs has gained prominence in veterinary medicine [1]. Nóbrega et al. (2019) [2] identified high expression of COX-2 and VEGF in 60 dogs with cutaneous HSA, suggesting that these may represent potential therapeutic targets. Further clinical studies are required to prove that their inhibition leads to antitumor effects.

Photodynamic therapy has also emerged, with limitations in surgical resection, either by extension or by several lesions. Rocha et al. (2019) [72] evaluated the use of this therapy in eight dogs with cutaneous HSAs and achieved complete remission in 90% of cases. As this is a new therapy, prospective studies with more animals are needed.

The association between actinic dermatosis and cutaneous hemangiosarcoma can be explored therapeutically using natural photoprotectors such as omega-3 fatty acids, vitamins, and polyphenols [73,74]. Omega-3 polyunsaturated fatty acids (AGPs) primarily comprise α-linoleic acid (ALA), docoHSAexaenoic acid (DHA), and eicosapentaenoic acid (EPA) and come directly from foods and/or dietary supplements enriched with omega-3 fish oil. Omega-3 AGPs exert anti-inflammatory effects. Studies involving its metabolites, such as resolverins and maresins, have revealed potent anti-inflammatory, analgesic, and antineoplastic activities, mainly by reducing the production of cytokines/chemokines derived from neoplastic cells (TNF-α, IL-6, CXCL10, and MCP-1) and CD11bþLy6G myeloid cells induced by tumor mediators and nociception. A decrease in ROS production and improvement in the phagocytic activity of macrophages were also observed in one study [75,76,77,78]. Treatment with maresin 1, a lipid mediator derived from omega-3 AGP, was associated with protection against UVB radiation in hairless mice. The results showed less inflammation and oxidative stress in the treated animals, which was related to the inhibition of edema, keratinocyte apoptosis, and the presence of mast cells. Furthermore, the lower production of metalloproteinases 9 (MMP9) and the consequent degradation of collagen fibers induced by radiation were attributed to the administration of maresin 1 [16].

Vitamins B3 and C and natural polyphenols, including cannabis derivatives, have been associated with the photoprotective activity of the skin [73,79,80,81]. A study that exposed rats to UVA and UVB radiation demonstrated the protective effect of phytocannabinoid-cannabidiol (CBD) on the lipid metabolism of keratinocytes, reducing 4-HNE and 8-isoPGF2α lipid peroxidation products. This experiment also demonstrated that CBD increased the transcriptional activity of Nrf2 and the expression of its inhibitor, Bach1, and prevented the UVA/UVB-induced increase in the expression of Nrf2 activators p21, p62, p38, and KAP1, and pro-inflammatory factors, such as NF-κB and TNF-α [80].

## 10. Prognostic

The biological behavior of non-visceral HSA varies according to the tumor’s location and degree of tissue invasion, with clinical staging considered the most important and well-established prognostic factor for cutaneous, subcutaneous, and muscular presentations.

After complete surgical resection, the prognosis for dogs with actinic cutaneous HSA and no metastatic spread (stage I) was considered favorable. In a retrospective study by Szivek et al. (2011) [11], dogs with cutaneous HSAs (with or without the presence of actinic alterations) treated with surgery alone achieved median overall survival ranging from 780 to 987 days, with 79%, 60%, and 44% of dogs alive in the first, second, and third years after diagnosis, respectively. However, in the same study, dogs of non-predisposed breeds diagnosed with non-actinic HSA had shorter survival times and higher metastatic rates than those of cutaneous actinic HSA and predisposed breeds (median survival of 1549–1570 days vs. 539–593 days, respectively).

In a study by Sviek et al. (2011) [11], locoregional recurrence was not associated with worse prognoses and was common in predisposed dog breeds with cutaneous HSA with actinic alterations, even when complete surgical margins were obtained. These results can be justified by the hypothesis of the “cancerization effect in the field,” which states that the long-term exposure of an anatomical region to a carcinogen (in this case, UV radiation) leads to the development of multiple microscopic neoplastic foci, which throughout the life of the animal can lead to the recurrence of tumors in the same anatomical region [11,82].

On the other hand, HSAs that arise or infiltrate subcutaneous and/or muscular tissues (stages II and III) have a worse prognosis [3,25,26,29]. In a study by Ward et al. (1994) [3], dogs with subcutaneous and muscular HSAs had significantly lower survival rates than those with HSAs confined to the dermis (172 and 307 days vs. 780 days, respectively). In the study by Sviek et al. (2011) [11], although subcutaneous invasion did not influence the survival time of affected dogs, the risk of metastases in these patients was considered twice as high as in those with tumors confined to the dermis.

Larger tumors have higher rates of incomplete resection, making adequate local disease control unfeasible. Therefore, tumor size has been investigated as a potential prognostic factor in both humans and dogs with cutaneous HSAs. Humans with lesions >5 cm in diameter have a worse prognosis than those with tumors <5 cm [83,84]. In turn, dogs with lesions >6 cm had significantly lower survival and disease-free times than those with lesions <4 cm [26]. However, Schultheiss (2004) [29] found no association between tumor size and prognosis.

Complete surgical excision of lesions has been associated with prolonged survival and better prognosis [3,26,29]. Shiu et al. (2011) [26] demonstrated that complete tumor excision in dogs with subcutaneous and muscular HSAs, mainly when associated with smaller tumor size (<4 cm) and absence of metastases at diagnosis, provided significantly higher survival than incomplete resection (399 vs. 130 days, respectively). These authors also hypothesized that complete resection of tumors is more critical in reducing metastatic risk than preventing local recurrences.

Some studies have evaluated the prognostic value of different immunohistochemical markers in canine HSA; however, their expression indices have not shown prognostic value [2,31,85]. Nóbrega et al. (2019) [2] conducted the first study in which the prognostic value of the expression of immunohistochemical biomarkers (FVIII, COX-2, VEGF, PCNA, and caspase-3) was evaluated exclusively in cutaneous HSA. However, the expression of these markers was not correlated with the survival of the 60 dogs evaluated.

Given the above, tumor staging should be considered as the main prognostic factor for canine cutaneous HSA, mainly considering the origin of the primary tumor and the degree of invasion. Other clinical–epidemiological characteristics that help define prognosis and can be evaluated together with staging include (1) the presence of actinic lesions, (2) predisposed breeds, (3) complete surgical resection, and (4) tumor size (Table 3). Further studies are needed to validate the prognostic value of these factors.

## 11. Prevention

Given the correlation between exposure to UV radiation and the development of this disease in dogs with lightly pigmented coats and those of predisposed breeds, it is possible to define recommendations to prevent the appearance and tumor recurrence in dogs known to be predisposed and healthy as well as in those with a previous history of neoplasm. Thus, tutors must be educated about the following preventive measures: (1) restriction of sun exposure, (2) increased sun protection of the skin using suitable sunscreens and protective clothing, and (3) constant monitoring of the animal for the emergence of new nodules.

However, it is essential to consider that in dogs with a history of chronic sun exposure or a previous history of actinic HSA, damage to the genetic material caused by UV radiation may have already occurred, and the reduction in sun exposure may not be more effective in preventing the appearance of lesions and tumor recurrence. This reinforces the need to implement these preventive measures early in a patient’s life, especially for those at greater risk. Prospective studies are needed to determine the effect of preventive measures in reducing the damage induced by solar radiation in dogs with actinic cutaneous HSA [11].

## 12. Splenic Hemangiosarcoma

### Etiology and Epidemiology

Neoplasms of vascular origin that affect the spleen are highly prevalent in dogs and have a high frequency of malignancy [1,86,87]. Given the prevalence of malignant tumors in the spleen, Johnson et al. (1989) [88] proposed the so-called “two-thirds rule”, defining that two-thirds of neoplasms diagnosed in the spleen are malignant, and two-thirds of these are hemangiosarcomas. This rule remains a reference in the literature [86,89,90,91,92,93] for determining the prevalence of splenic HSA, the most evident neoplasm in the spleen of dogs.

Recent studies [18,29,94] suggest that the cellular origin of HSA is derived from bone marrow pluripotent cells in the pre-differentiation phase, which are able to migrate to sites of vascularization and undergo neoplastic transformation [5], which may explain the high prevalence of HSA in the spleen of dogs; however, its origin is still not fully understood [1,15,86,87].

Recent studies [95,96,97] suggest that infection with agents such as Bartonella spp. may correlate with the development of HSA in dogs because chronic infection exposure to these agents induces angiogenesis and chronic inflammation, which are factors that are fundamental in the proliferation and development of vascular tumors such as HSA.

Epidemiological factors vary according to the different studies and observed populations [5,86,87,89]; therefore, there are reports of splenic HSA in dogs of many breeds, including mixed breeds, of ages between 6 and 17 years old. However, a greater predisposition was observed in males [97]. Reproductive status is a risk factor, and it has been observed that castrated animals are more likely to develop the disease [92,98,99,100,101,102], regardless of the period of life in which the animal was castrated [102]. A greater predisposition is also observed in large dogs with an average weight of >25 kg [88,92,98,100,101]; Golden Retriever, Labrador Retriever, German Shepherd, Boxer, Cocker Spaniel, and mixed breeds are the most predisposed [88,92,100,101,103,104].

## 13. Clinical Manifestation and Biological Behavior

The clinical manifestation of signs in dogs with splenic HSA varies and may depend on several factors, such as tumor size, presence or absence of metastasis, and mass rupture [1,5,15,86,87]. Splenic HSA is considered the primary cause of non-traumatic hemoperitoneum in dogs [103].

The clinical history is usually composed of nonspecific signs, such as anorexia, lethargy, weakness, cachexia, dyspnea, syncope, pallor, increased capillary refill time, dehydration, peripheral hypotension, cold limbs, emesis, diarrhea, and hyperthermia; these symptoms appear acutely in cases of rupture or chronically and are followed by local signs, such as abdominal distention, splenomegaly, abdominal pain on palpation, and abdominal effusion/hemoperitoneum. Tumor rupture can cause hypovolemic shock, arrhythmias, respiratory arrest, and even sudden death [1,5,15,86,87,89,93,105]. Despite the wide variety of clinical signs associated with HSA, this tumor can also be found during routine examinations [100].

Frequently encountered laboratory abnormalities include regenerative anemia and thrombocytopenia [1,5,15,86,87,89,93,106]. Disseminated intravascular coagulation, where factors such as increased prothrombin time and activated partial thromboplastin, increased levels of D-dimers, hypofibrinogenemia, decreased antithrombin activity, and fragmentation of red cells may be present [93], while other changes in leukogram and serum biochemistry are not frequently observed. Hypoalbuminemia and increased liver enzymes have also been identified [1,107].

One or multiple splenic nodules can be identified in patients with HSA. Because of its highly metastatic nature, it is assumed that the main mechanisms of metastasis of splenic HSA are via the hematogenous route or by seeding cells in organs of the abdominal cavity due to tumor rupture [1,5,15,86,87,93], with the liver, omentum, peritoneum, lung, right atrium, and brain being the most common sites of metastasis [108], and less commonly in bones and skeletal muscle, as reported by Carloni et al. (2019) [44].

## 14. Diagnosis and Staging

Diagnosis and staging of splenic injuries usually involve performing abdominal ultrasonography and three-view chest radiography. In cases where there is free abdominal fluid, analysis of cavity fluid (if there is enough) and paired evaluation of hematocrit are performed to confirm hemorrhage [1,5,15,86,87,109].

Hematological tests, such as complete blood count and serum biochemistry of renal and hepatic profiles, are recommended, but the results cannot characterize malignancy. The patient’s coagulation profile should be evaluated when possible because dogs with splenic HSA are predisposed to DIC [5,86].

Radiographic chest evaluation in three projections aims to search for metastasis, as well as cardiological evaluation and observation of the presence or absence of thoracic lymph node enlargement [1,5,15,86]. Hammer et al. (1993) [110] observed that the miliary radiographic pattern was the most common in pulmonary metastasis, representing 73% of the findings. Well-circumscribed nodules are less common and lesions are consistently observed in a non-isolated manner. Cardiological evaluation is commonly performed for anesthetic planning but also because of the need to investigate arrhythmias that patients with splenic HSA may present [105]. Approximately 47% of metastatic or concomitant HSA lesions in the heart cause radiographic changes in the cardiac silhouette, with echocardiographic evaluation being the choice for confirming possible cardiac neoformations [1,5,15,28].

Ultrasonographic evaluation of the abdominal cavity aims to look for peritoneal effusion, presence and characterization of nodules in the spleen and liver, and search for metastasis in adjacent organs [109,111,112]. The characterization of splenic tumors is based on their size, echogenicity, and number of nodules, as well as the presence or absence of cavitation. Cystic, large, or ruptured structures should be investigated, and splenectomy is recommended to reduce the risk of death. Yankin et al. (2019) [109] evaluated that 20% of splenic tumors between 1 and 2 cm are more likely to be clinically relevant compared to lesions smaller than 1 cm, as well as the presence of abdominal effusion and “target” lesions—a finding that is usually rounded or oval with different concentric echogenicities, between the periphery and the center, or in concentric layers—are also more likely to be clinically relevant. Levinson et al. (2009) [113] and Davies and Taylor (2020) [92] reported that hemoperitoneum associated with splenic injury in dogs is correlated with a high probability of malignancy. However, approximately 40% of dogs with histologically confirmed splenic hematoma had hemorrhagic peritoneal effusion [114].

Although splenic and hepatic B-mode ultrasound evaluation is considered a sensitive method for detecting lesions in these tissues, it is somewhat sensitive for differentiating or detecting malignancies [115,116]. Ultrasonographic features frequently observed in splenic and hepatic hemangiosarcomas include the presence of complex echogenic neoformation, multiple and small cystic areas producing posterior acoustic reinforcement, marked alteration in the surface of the organ and its contour, with several nodules distributed throughout the parenchyma (1–10 cm), and anechoic (cystic) or hyperechogenic solid lesions [115,116]. Watson et al. (2011) [117] highlighted the difficulty in differentiating malignant and benign structures using ultrasound examination alone.

Microbubble-enhanced ultrasound (CEUS) can be used as a complementary method to the B-mode for differentiating malignant lesions in the liver and spleen of dogs. Using this technique, malignant splenic and hepatic lesions present hypoechogenicity in the filling of the ultrasound contrast compared with the adjacent normal tissue. Specifically, for hemangiosarcoma, in addition to the hypoechogenicity observed in all filling phases, there was a lower wash-out rate, faster ultrasound contrast output, and the presence of tortuous and aberrant vessels. However, the presence of hematomas can be difficult to differentiate from hemangiosarcomas using CEUS [118,119,120].

Elastography, which evaluates tissue stiffness using ultrasound, can also enhance B-mode study. Maronezi et al. (2022) [112] achieved an accuracy of 97% for differentiating malignant lesions, and the higher occurrence of malignancy was attributed to the diagnosis of hemangiosarcoma. Malignant lesions are more rigid than benign ones, verifying this characteristic in the elastogram and values greater than 2.6 m/s for shear velocity. This method can corroborate the CEUS data since it allows differentiation, addressing a limitation mentioned in the evaluation of the contrast-enhanced ultrasound technique in splenic lesions.

Advanced imaging tests, such as computed tomography, PET-CT, and magnetic resonance imaging, can be used to perform more accurate tumor staging, planning, and restaging to select patients to proceed with therapy [1,5,86,121].

A triphasic tomographic study allows the differentiation of splenic neoformations through the analysis of the enhancement pattern by intravenously administered contrast. Splenic HSA has been associated with a remarkably heterogeneous pattern in the arterial and portal venous phases and a poor and homogeneous enhancement pattern in all phases [121].

The heterogeneous enhancement pattern is significant in several veterinary studies, where hemangiosarcomas in different locations present an essential hypervascular component in the arterial phase (predominantly peripheral) and heterogeneous enhancement foci in the portal venous and late post-contrast phases, with major slow and continuous expansion, usually followed by blood attenuation or vessel signal intensity [37,121,122,123]. Similar findings have been reported in human studies [124,125].

Thus, computed tomography in veterinary medicine is a minimally invasive examination for these patients, capable of precisely locating the lesion, and is useful for planning and surgical therapeutic approaches. Tomography is more sensitive than chest radiography for the early detection of pulmonary micrometastases and other distant metastases [39,44]. It also differentiates tissues by their attenuation value, which is related to the density and chemical composition of the tissue, often making it possible to distinguish between non-malignant and malignant lesions. It also presents high diagnostic accuracy in cases of hemangiosarcoma, in which case sampling can avoid hypervascular and cystic structures and reduce complication rates due to hemorrhage [117,126,127,128].

In a study using contrast-enhanced PET-CT, it was observed that the whole-body evaluation is an excellent tool in tumor staging due to the ability to identify small foci of metastasis in different organs [129]. This diagnostic modality can contribute to adequate tumor staging, mainly for visualizing metastatic lesions in the central nervous system [130].

Although computed tomography provides better spatial resolution, especially in the location of lesions, magnetic resonance imaging is a non-invasive imaging modality with promising potential for evaluating lesions with a soft tissue component, with better contrast and the ability to discriminate subtle differences between tissues. Splenic hemangiosarcoma can be characterized by an intermediate signal intensity with the splenic parenchyma on TW1-weighted sequences, hyperintensity on TW2, lower signal with slight peripheral enhancement on post-contrast TW1, hyperintensity on T2, isointensity on T1, and no signal of enhancement using contrast. Low signal intensity on TW1 and TW2 sequences usually reflects the presence of hemorrhagic necrosis or siderotic nodules. The hyperintense region in these sequences may correspond to a late subacute hemorrhage. Thus, magnetic resonance imaging can increase the diagnostic accuracy of lesions [126,127]; however, this test does not accurately identify pulmonary metastases.

Fine-needle aspiration cytology has little diagnostic efficiency because of the intense blood contamination associated with material collection [1,5,15,86,87,109], in addition to the risks of metastasis due to seeding and hemorrhage due to rupture of the tumor capsule. The positive correlation between the diagnosis made by aspiration cytology and tru-cut biopsy in splenic nodules was only 51.4%, with 40% of the total disagreement between the results of these tests [117].

The evaluation of liver metastasis by trans-surgical biopsy has been reported [91,131,132] to assess the involvement of the organ at an early stage, with the observation that collecting samples from livers with normal morphology is unnecessary because only 2.5% of the animals without liver morphological alteration had metastatic disease at the time of surgery.

The definitive diagnosis of splenic HSA involves histopathological examination. Macroscopically, these nodules are usually white, reddish, or purple with a soft, gelatinous, and friable consistency, commonly filled with blood and areas of necrosis [1,87,100], and the omentum may have adhered because of possible ruptures. Because of the large dimensions that splenic HSA can reach, it is common to send only a few tumor fragments for analysis. However, extensive areas of hematoma can occur in large splenic tumors, such as HSA [114], and the absence of malignancy characteristics can indicate a false negative result if insufficient sampling or unrepresentative areas are used. At least five tumor fragments should be analyzed for higher accuracy of diagnosis [133].

The entire spleen is recommended to be sent for analysis so the pathologist can choose viable fragments for examination. HSA usually comprises well-differentiated lesions, but a small portion may be poorly differentiated or solid, where the cell morphology is not characteristic. Immunohistochemical evaluation is recommended for the differential diagnosis of other sarcomas that may affect the spleen, such as stromal, histiocytic, and hemophagic histiocytic sarcoma, as shown in Figure 6 [1,15,87,134].

Although a definitive diagnosis is made by histopathological examination, several studies have sought non-invasive diagnostic methods for characterizing the malignancy of splenic tumors, such as the evaluation of serum VEGF, tumor DNA, and microRNA profiles [135,136,137,138]. Frenz et al. (2014) [135] observed that circulating VEGF levels were significantly higher in animals with splenic lesions, but there was no difference between benign and malignant lesions. Similarly, Favaro et al. (2022) [138] observed that serum tumor DNA was significantly higher in patients with splenic HSA than in those with benign lesions. However, excision did not significantly reduce the fraction of circulating tumor DNA, and the authors considered maintaining the serum levels of tumor DNA due to the presence of metastases. Grimes et al. (2016; 2021) [136,137] established the significance of comparing the expression of circulating microRNA profiles between animals with splenic lesions and healthy animals, not being able to characterize malignancy by the test. Although the findings are not related to tumor staging, the authors consider the use of analyses as a noninvasive diagnostic tool for splenic masses, requiring further studies to validate their ability to apply them in clinical practice.

Recent advances in genetic sequencing technology and bioinformatics have allowed for constant improvements in the study of cancer, clarifying questions about genetic mapping and the molecular properties that they present and favoring the development of target-specific therapeutic protocols [139]. Studies [19,140] have identified mutations in signaling pathways in canine HSA and human angiosarcomas, implying that drugs that inhibit these activation pathways may be used to treat HSA. Although further studies are needed, the popularization of the technique should reflect a greater frequency of genetic mapping of tumors, contributing to the formation of epidemiological data on HSA mutations in dogs. This type of study is also beneficial in comparative oncology, given the remarkable similarity between canine HSA and human angiosarcomas and the higher frequency of the former than the latter [19,140,141,142].

The clinical staging of canine visceral HSA considers tumor size, as proposed by Mullin and Clifford (2020) [1], as shown in Table 4. However, Wendelburg et al. (2015) [33] and Bray et al. (2017) [143] presented a simplified staging system (Table 5) that excluded tumor size, as this does not correlate with prognosis. These cavitary lesions do not necessarily present with a more aggressive course.

## 15. Treatment

The treatment of choice for HSA is surgical, not different from splenic HSA, and metastasectomy of apparent lesions that are observed during laparotomy can be performed using the same procedure, often performed on an emergency basis due to hemoperitoneum caused by tumor rupture [1,5,15,86,87,144,145,146].

Owing to the high metastatic capacity of splenic HSA, the association of tumor excision with adjuvant therapies is recommended to increase survival time, disease-free time, and better quality of life for the patient. A study in the United Kingdom [147] revealed that less than 50% of veterinarians indicated the maximum tolerated dose chemotherapy protocol (QMDT) as adjuvant therapy for splenic HSA for reasons such as ineffectiveness, high cost, and lack of qualified professionals. Batschinski et al. (2018) [146] observed a statistical difference in the mean survival time when surgery was associated with QMDT, with values of 66 days for surgery alone and 274 days for combination chemotherapy. Several QMDT protocols are available in the literature, including monotherapy protocols with doxorubicin, carboplatin, ifosfamide, and epirubicin, as well as combined protocols such as vincristine, doxorubicin, and cyclophosphamide (VAC) and doxorubicin and cyclophosphamide (AC); despite the wide range of available protocols, the mean survival time did not vary significantly between them, ranging from 130 to 170 days [1,5,15,23,25,26,32,69,71,86,87,146,148,149,150,151,152,153]. The choice of chemotherapy protocol is influenced by the presence or absence of heart disease, which restricts the use of doxorubicin in these patients.

Studies have evaluated the association of chemotherapy with the metronomic dose (QM) or in an isolated protocol, which consists of dividing the maximum tolerated dose into daily fractions of lower doses to reduce the side effects that QMDT can cause [33,71,152,154]. For patients with TI or TII clinical staging, the average survival time did not differ between patients who received QM or QMDT, either alone or in combination. Marconato et al. (2019) [152] speculated that for patients in clinical stage TIII, with the presence of metastasis, the anti-angiogenic effect of QM may not be practical due to the rapid growth rate of the tumor. Thus, QMDT is a viable alternative for patients in the TI and TII stages. Further studies are needed to reinforce the effectiveness of the combination of QMDT and QM therapies in increasing mean survival time.

Several studies have used targeted therapies. Gardner et al. (2015) [155] administered toceranib after completing the QMDT protocol with doxorubicin but did not observe an increase in the mean survival time compared to animals that received only the chemotherapeutic agent. Borgatti et al. (2017; 2020) [156,157] evaluated the efficacy of an angiotoxic drug with a targeted therapeutic effect against sarcomas and tumor neovascularization in dogs (eBAT), administered to dogs with clinical stages TI and TII after splenectomy and prior to the QMDT protocol with doxorubicin. The results showed an increase from 40% to 70% in the number of patients who survived to 6 months in relation to the group that received only QMDT with doxorubicin. Six dogs survived more than 450 days; this may be an adjuvant therapeutic option in dogs diagnosed with splenic HSA. Another drug that may have beneficial effects in the treatment of splenic HSA is thalidomide because of its potential VEGF-inhibitory effect and may be associated with QMDT and QM protocols [5,71,143,158]. Although more studies are needed to consolidate the effects of thalidomide on the mean survival time, the results obtained by Bray et al. (2017) [143] demonstrated that 33% of patients receiving thalidomide had a survival time greater than one year, overlapping the average survival time of 170 days observed in the QM and QMDT studies.

Recent studies [159,160,161] reported high expression of beta-adrenergic receptors in several types of cancer, including angiosarcomas and hemangiosarcomas. They observed that such receptors, in the signaling of catecholamines such as epinephrine and norepinephrine, are responsible for activating pathways of cell growth and proliferation, angiogenesis, and inflammation, as well as dysregulation of cellular mobilization pathways. These findings paved the way for several clinical studies and case reports [162,163,164,165] that used propranolol, a drug of easy access, which has been studied for decades in several species; minimal side effects were reported when used as a single agent or associated with QMDT, QM, and radiation protocols, with an increase in disease-free time, an increase in the average survival time, and regression of lesions and metastatic foci. However, the authors state that, although the findings of these studies are encouraging in terms of improving the prognosis of patients affected by angiosarcomas and hemangiosarcomas, large-scale studies are needed to integrate propranolol as part of the therapeutic regimen for vascular sarcomas [164].

Alternative therapies using natural products are a new source of research for treating HSAs, such as herbal extracts (Yunnan Baiyao) and mushrooms (Coriolus versicolor) have been reported [86,166,167,168,169]. Brown and Reetz (2012) [166] reported that 100 mg/kg/day of Coriolus versicolor led to a mean survival time of 199 days, which was slightly higher than that reported in studies with QMDT and QM and may be an alternative in places where access to antineoplastic chemotherapy is difficult. Gedney et al. (2022) [169] reported that Coriolus versicolor with QMDT with doxorubicin did not increase the mean survival time and that females had a shorter mean survival time than males who received the treatment alone. Wirth et al. (2014) [167] reported that the administration of Yunnan Baiyao in HSA cell culture led to cell death via the induction of apoptosis via caspases. Ciepluch et al. (2017) [168] observed that animals receiving Yunnan Baiyao had longer disease-free intervals than animals receiving only chemotherapy or blood transfusion. However, because of the small number of animals that received the herbal extract, further studies should be conducted to verify the effectiveness of Yunnan Baiyao in treating HSA in dogs.

Target-specific molecular therapies have been studied for canine HSA because of their high incidence compared with human angiosarcoma. More specific drugs that inhibit PI3K/mTOR/Akt and MAPK/ERK activation pathways [140,170] induce cell apoptosis and inhibit cell proliferation and migration [171]. Although the studies were in vitro, the results of targeted therapies have shown promising results.

### Prognosis

The prognosis of splenic HSA in dogs is unfavorable [1,15,86,87,144,172,173,174,175] because of the high metastatic capacity of the tumor and the high rate of metastasis at the time of diagnosis. The prognostic factors with the most significant impact should be evaluated as follows:Clinical staging: patients with TI and TII staging have a longer average survival time than patients with TIII staging; the possibility of administering a chemotherapy or adjuvant protocol increases the survival time of patients who are only subjected to splenectomy.Anemia, thrombocytopenia, and the need for blood transfusion at the time of surgery are considered unfavorable factors for the prognosis of HSA in dogs [89,106,145,168] as they are associated with more advanced clinical staging and worse clinical status of the patient.Mitotic index (MI) of less than 11 is associated with a higher survival rate and is correlated with a slower tumor growth rate [176].Immunohistochemical evaluation: a low expression of Ki67 (below 16%) and PSMA is associated with better survival, while CLAUDIN 5 expression is associated with an increased risk of metastasis [175].

Thus, the combination of splenectomy and chemotherapy remains the therapy of choice to enable a better prognosis in dogs affected by splenic HSA, with a mean survival time of 170 days and clinical staging as the predominant factor, as shown in Table 6.

## 16. Cardiac Hemangiosarcoma

### 16.1. Etiology and Epidemiology

Cardiac neoplasms are uncommon in canines and even less common in felines, with an incidence varying between 0.17 and 0.19% of tumors, and less than 0.03% of tumors, respectively [15,177,178]. Hemangiosarcoma (HSA) is the main cardiac tumor in dogs [1,5,15,28,87,177,178].

The cardiac location is the third most frequent site of HSA according to the national literature [15] or the second most frequent site according to the international literature [1]. Cardiac tumors are primary tumors in approximately 84% of cases [15,177]; however, cardiac HSA can also occur as a metastatic disease [87].

The epidemiological profile of greater involvement by cardiac HSA mainly includes German Shepherd and Golden Retriever breeds, with an increased incidence in the Maltese and Dachshund breeds [15,87,177,179,180,181]. Adult animals between 7 and 15 years of age are the most affected, although there are reports of young animals up to 2 years of age [15,87,177,179,180].

Sexual predisposition is unproven [15,180], but reproductive status has been linked to the development of HSA, with a significantly higher incidence in castrated females and slightly higher in castrated males [87,177]. The influence of castration has been proven to be related to splenic HSA, but there is no proof of its relationship with cardiac HSA [102].

Despite the low incidence, it is essential to know cardiac neoplasms because, when present, they require rapid recognition and intervention to guarantee the best possible prognosis for the patient [15].

### 16.2. Clinical Manifestation and Biological Behavior

HSA most commonly develops in the right atrium and usually as solitary tumors; however, multiple nodulation can occur within the atrium, auricle, and adjacent tissues [1,5,15,87,178,179,180]. Less common locations include the right ventricle, left-sided heart chambers, and intraventricular septum [5,180,182,183,184,185].

Cardiac HSA presents with local and distant aggressive behavior. Local invasion and spread to adjacent tissues are common, which can lead to rupture of the atrial wall with the development of pericardial effusion and cardiac tamponade [15].

Metastatic potential is equally high, matching splenic HSA, and occurs early [1,15,180]. The main routes of dissemination are hematogenous and intracavitary and are usually related to tumor rupture [1]. The target organs are the lungs, liver, and spleen; less frequently, the kidneys, mesentery, intestines, omentum, central nervous system, adrenal glands, peritoneum, visceral lymph nodes, and diaphragm [15,180,186]. Metastases to the pericardium and other cardiac chambers have also been observed [15,87,180], with reports of metastases to multiple organs [15].

It is estimated that approximately 62% of cases progress to pulmonary dissemination at the time of the first consultation, and metastases are observed in post-mortem examination in 75% of cases [15]. Simultaneous manifestations of cardiac and splenic HSA have also been described, with variable incidences [1,5,108].

Clinical signs are related to the location and size of the mass, especially the presence of pericardial effusion and cardiac tamponade. Fluid accumulation in the pericardium occurs due to the hemorrhagic characteristics of the neoplasm, leading to an increase in intrapericardial pressure. Hypertension interferes with atrial and ventricular filling and consequently decreases cardiac output and arterial pressure, which, when reduced, causes neurohumoral, adrenergic, and renal responses [5,15,87,178,187,188].

Cardiac/pericardial neoplasms account for more than 60% of pericardial effusion cases in dogs [178]. However, it is important to emphasize that the absence of pericardial effusion does not rule out the presence of cardiac mass [1,183].

The clinical signs of affected animals include lethargy, apathy, anorexia, weakness, weight loss, emesis, exercise intolerance, dyspnea, cough, syncope, and collapse [15,28,178,179,180,183,188,189,190,191]. Physical examination revealed muffled cardiac and pulmonary auscultation, arrhythmias, weak pulse, jugular distention, pale mucous membranes, abdominal distension, dehydration, tachypnea, and tachycardia [5,15,28,179,180,189,190]. Patients with atrial rupture present a critical condition with rapid progression, and sudden death may occur [1,15].

Intracardiac neoformations can occur with different repercussions, with cardiogenic pulmonary edema, manifestation of murmur, or remain asymptomatic or with mild symptoms of exercise intolerance, with acute evolution and sudden death [178,182,184,185].

HSA can also occur in an extracardiac intrathoracic location, mainly in large vessels and in the pericardium, with a similar clinical manifestation but with punctual differences, such as the manifestation of a murmur [178,190,192,193,194,195]. Atypical symptomatology, such as neurological conditions, can also occur in relation to the presence of distant metastases [186].

### 16.3. Diagnosis and Staging

The diagnosis of cardiac HSA can be divided into two fundamental points: detection and location of the mass and diagnostic confirmation of the histological type. However, many cases do not reach the second stage, remaining only as a presumptive diagnosis of HSA.

The suspicion of cardiac neoplasm was established based on the joint evaluation of the patient’s history and clinical signs, physical examination findings, and complementary examinations. In asymptomatic cases, usually without pericardial effusion, the diagnosis may be accidental during routine examinations or investigation of concomitant diseases [5,15,178].

Fluid cytology tends to be the first test performed since many animals are diagnosed with pericardial effusion, requiring immediate stabilization through pericardiocentesis or thoracentesis [15,181].

Fluid analysis is effective in classifying the effusion as hemorrhagic, transudative, or infectious, but differentiation between neoplastic and non-neoplastic samples is not possible in most cases [15,178,181]. Confirmatory diagnosis of HSA is even more difficult due to the low exfoliation of this cell type, in addition to the high hemodilution present in the samples [1,5,15,181]. However, cases of HSA more commonly occur with hemorrhagic effusions [5,15,178,181]. The biochemical evaluation of the effusion is not effective in determining whether it is neoplastic or not [196].

Chest X-rays are always indicated in three projections (left side to side, right side to side, and ventrodorsal) as an effective test for the detection of large-volume pericardial effusion and cardiac volume increases [1,15,178,189]. In the presence of pericardial effusion, examination shows an increase in the cardiac silhouette, with a globe-shaped heart, without specific enlargement of the chambers [1,5,15,28]. It is estimated that the cardiac silhouette is altered in approximately 47% of cardiac HSA [28]. The radiographic appearance of metastasized lungs is often described as a nodular to interstitial coalescing miliary pattern [1].

Echocardiography is most commonly used to confirm the presence of cardiac masses, allowing not only the visualization of the mass in most cases but also the diagnosis of initial pericardial effusions [1,15,28,179,183,189,197]. The test allows the differentiation of masses in the right atrium from those located in the cardiac base with high sensitivity and specificity [5,197]. It is important to emphasize, however, that false-positive and false-negative results can occur, and that the absence of a visible tumor does not exclude the possibility of cardiac HSA [5,28,178,184,189,198].

Electrocardiography may show alterations such as premature ventricular contractions and ventricular tachycardia. Other changes include atrial fibrillation, premature atrial complexes, atrial tachycardia, and reduced amplitude of the QRS complex [1,5,15,178,189]. Pericardial effusion can also cause suppression of millivoltage and/or electrical alternation [1,15,28].

Advanced imaging methods, such as computed tomography and magnetic resonance imaging, can also be used, with the aim of identifying the tumor, in addition to allowing greater precision in its dimensions [5,15,178,199]. They are mainly indicated when echocardiographic examination does not confirm the presence of a mass and/or the involvement of adjacent structures. Computed tomography and magnetic resonance imaging represent the examination of choice in the search for abdominal metastases, while CT stands out in the diagnosis of thoracic metastases as well as in surgical planning [1,5,15,199].

Whole-body contrast-enhanced computed tomography is indicated and is an important diagnostic aid in the identification and location of heart lesions (with three-dimensional visualization of the mass), as well as the evaluation of surrounding tissues and vascular structures, contributing to therapeutic surgical planning. It allows the characterization of small pericardial and pleural effusions, in addition to assessing the presence of pulmonary and distant metastases, especially those involving abdominal organs, which is of crucial importance for establishing prognosis [200].

Laboratory tests, including blood count, serum biochemistry, and coagulograms, should always be performed to assess the general condition of the patient and identify comorbidities. The main hemogram findings related to cardiac HSA are leukocytosis with a left shift and regenerative or non-regenerative anemia in addition to the presence of schistocytes and acanthocytes, alterations associated with microangiopathic damage, vasculitis, and acute hemorrhage [1,15,179,180]. Thrombocytopenia and coagulation disorders have been previously described [179].

Abdominal ultrasonography is indicated for the purpose of staging and researching metastasis, and the lesions, when present, are observed with a heterogeneous appearance and may have mixed echogenicity and areas of cavitation [1,15,179]. The staging of cardiac HSA is similar to that of other locations [1,15].

Tumor cytology can also be performed, being less invasive compared to biopsy, but the diagnosis will not always be reached because of low cellular exfoliation and high hemodilution [1,5,15,178,201]. When significant samples are obtained, the tumor appears to be similar to other locations, composed of fusiform, pleomorphic cells, and hyperchromatic nuclei, with intense malignancy criteria and may give rise to vascular spaces filled with blood [1,15,87].

Therefore, diagnostic confirmation is performed through histopathological examination; however, this represents a challenge in cases of cardiac tumors, as it is not always possible to safely collect the material [5,15,178]. Thoracotomy and thoracoscopy techniques are described for performing biopsy, but the diagnosis is often confirmed only in postmortem evaluation [15,178]. Macroscopically, cardiac HSA presents as an irregular, reddish mass, or, in initial cases, only the hyperemic aspect associated with atrial wall thickening may be present [87,180,183,184].

Poorly differentiated tumors may remain inconclusive even after histopathological evaluation, and in these cases, immunohistochemistry is indicated, especially with the markers vimentin, CD31, and von Willebrand factor [15,87,180].

Different biomarkers have been evaluated in patients with cardiac HSA. Cardiac troponins are markers of myocardial injury with satisfactory sensitivity and specificity for myocardial ischemia and necrosis [202,203,204]. The plasma concentration of cardiac troponin I was evaluated, with a higher median observed in dogs with cardiac HSA than in those with pericardial effusion unrelated to HSA and in dogs with non-cardiac HSA, demonstrating its functionality as a diagnostic tool [5,178,202,205].

The main differential diagnoses for cardiac masses and effusions include other neoplastic types, such as chemodectomas, ectopic thyroid or parathyroid carcinomas, lymphoma, mesothelioma, fibrosarcoma, thymoma, or metastatic neoplasms, as well as non-neoplastic alterations, such as traumatic or infectious pericarditis, left atrial rupture, peritoneum-pericardial diaphragmatic hernia, pericardial cyst, and uremic pericarditis [15,178].

### 16.4. Treatment

Many patients are euthanized because of the difficulty in establishing an effective treatment, the associated clinical complications, and the highly aggressive behavior of this type of tumor [1,15,183,206].

Patients complaining of pericardial effusion should undergo pericardiocentesis as the first intervention to allow for patient stabilization in addition to assisting in the diagnostic investigation, especially in cases that progress to metastasis in the pericardium. However, drainage of pericardial effusion has palliative purposes, and in the absence of definitive treatment, it becomes recurrent [15,178,187,191].

Surgical excision remains the mainstay of definitive treatment for cases of HSA without established metastasis [1,5,15,28,178,206]; however, its use is rarely possible due to the complexity and risks associated with surgical interventions in this region [1,178].

Small masses in the right atrium or pericardium require a more favorable surgical approach and can be partially removed through right thoracotomy or middle sternotomy [1,5,15,179]. Thoracoscopy has also been described as a viable alternative for resectioning masses in the right atrium and auricle with satisfactory results [5,207,208]. Pericardial graft techniques for reconstruction after extensive tumor resection have already been described in the literature with satisfactory results; however, this maneuver is not routinely used [1,5,28,209].

When there is no possibility of direct intervention and/or when the formation of pericardial effusion recurs, subphrenic total or partial pericardiectomy can be performed [1,5,15]. Access can be obtained by left or right lateral thoracotomy, median sternotomy, “H” sternotomy, or thoracoscopy [5,15,187]. Thoracoscopy results in less postoperative pain and lower morbidity; however, it has a high cost related to the equipment, in addition to the need for an experienced operator [15,28,210].

Pericardiectomy minimizes clinical signs and prevents the recurrence of pericardial effusion and cardiac tamponade but does not increase tumor-related survival [1,15,28,178,187], except for masses located in the pericardium, which is considered a rare location [190,195]. Complications associated with this procedure include arrhythmias [15,187].

Chemotherapy is indicated in all cases of cardiac HSA, considering the highly aggressive nature of the disease [1,15,178]. Its use associated with surgical intervention enables more remarkable survival [15,107,178,179,180], while its use as a single therapy and its actual effectiveness have not been proven [15].

Doxorubicin is a chemotherapeutic agent for adjuvant or neoadjuvant use [1,107]. The protocols described include its use as a single agent, as well as in combination with other drugs in the VAC protocol (doxorubicin, vincristine, and cyclophosphamide) or a single combination of cyclophosphamide, methotrexate, and dacarbazine [1,15,32,178,179].

Radiotherapy is rarely described in cases of HSA; however, the use of radiation has already been evaluated, showing benefits in cardiac HSA, reducing the frequency of cardiac tamponade in cases of recurrent effusion, suggesting its use as a single therapy or in combination with chemotherapy [5,211].

Additional therapies have been studied to increase the chance of disease control. Anti-angiogenic therapies are recommended based on the vascular characteristics of neoplasms [1,154,206]. The main form of anti-angiogenic therapy is metronomic chemotherapy, with cyclophosphamide associated or not with piroxicam, although its effectiveness in cardiac HAS is still controversial [1,154,178].

Molecularly targeted therapies are also extremely promising, considering that PDGF, VEGF, and stem cell factor receptor expression have already been identified in canine HSA [1]. The use of masitinib, imatinib, and dasatinib has already been evaluated in vitro, while the effectiveness of toceranib in vivo in cases of splenic HSA remains unproven [1].

Yunnan Baiyao is an herb widely used in Chinese medicine, with anti-inflammatory, pro-homeostatic, analgesic, and healing properties, and has been used in veterinary medicine as an anti-hemorrhagic agent for neoplastic and non-neoplastic conditions, including HSA [1]. The use of Yunnan Baiyao in dogs with cardiac HSA has been proven to be safe; however, it did not show improvements in the control of clinical signs or increased survival [212].

### 16.5. Prognosis

Cardiac HSA has an unfavorable prognosis [5,15,206] and high mortality because of changes in the neoplasm itself or metastatic disease [15,28,180]. The average survival in treated animals ranges from 16 days to 4 months in different studies [15,28,178], with a few cases exceeding 4 months of survival [209]. When treated only by surgical resection, the average survival ranges from 45 days to 5 months, with a more prolonged survival associated with chemotherapy, totaling approximately 164 days [15].

## 17. Conclusions

Canine HAS is a complex disease and even with the several paper published in the last decades, a limited number of prognostic markers are available and new treatment options are necessary. Canine HAS is a highly metastatic disease, and new therapeutical options are necessary for patients with advanced disease.

## Figures and Tables

**Figure 1 cancers-15-02025-f001:**
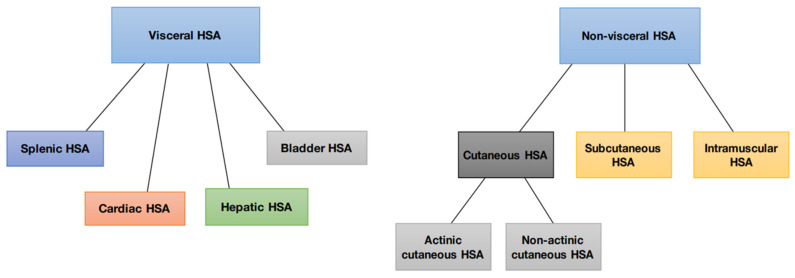
Schematic representation of the division of canine hemangiosarcoma into visceral and non-visceral.

**Figure 2 cancers-15-02025-f002:**
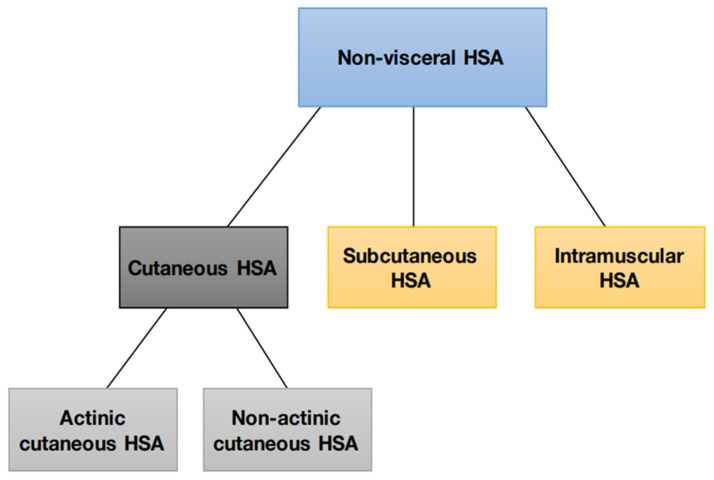
Most reported non-visceral HSA subtypes in dogs.

**Figure 3 cancers-15-02025-f003:**
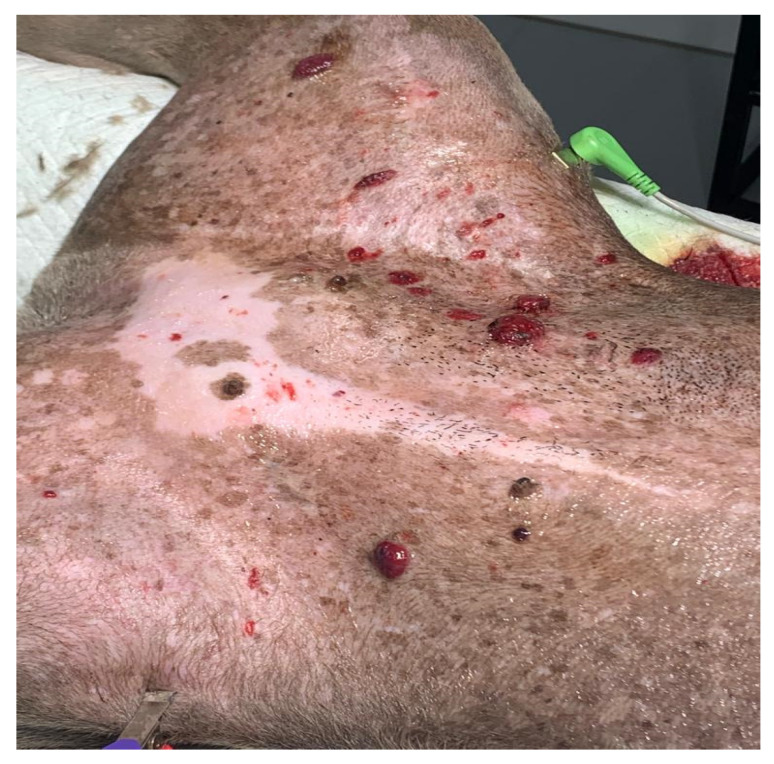
Multiple actinic lesions in a Pit Bull dog that had a history of chronic sun exposure.

**Figure 4 cancers-15-02025-f004:**
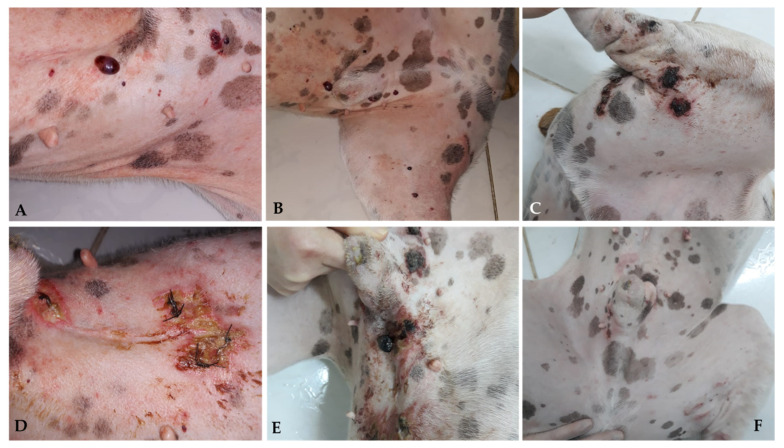
Canine patient diagnosed with multiple cutaneous HSA nodules; ECT with systemic BLM was administered. Multiple skin lesions are observed in the ventral abdomen and medial surface of the pelvic limb (**A**,**B**). Crust and ulceration 7 days after ECT (**C**–**E**). Complete remission 30 days after ECT with areas of scar tissue (**F**).

**Figure 5 cancers-15-02025-f005:**
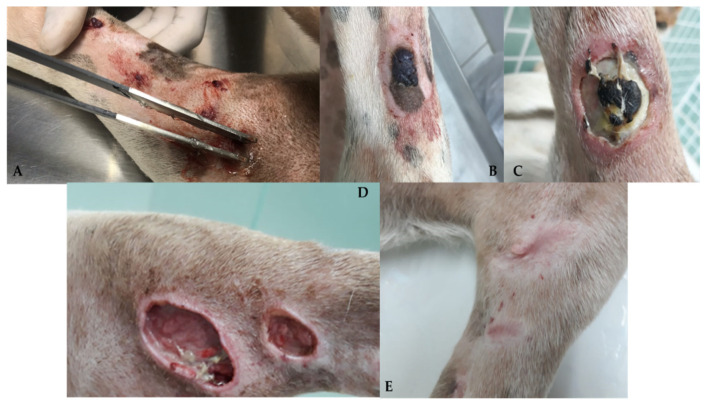
Canine patient diagnosed with multiple cutaneous HSA who underwent ECT with systemic BLM. Multiple skin lesions are observed in the region of the pelvic limb (**A**). Crust and ulceration 7 days after ECT (**B**,**C**). Necrosis and tissue loss 15 days after ECT (**D**). Complete remission 30 days after ECT with areas of scar tissue (**E**).

**Figure 6 cancers-15-02025-f006:**
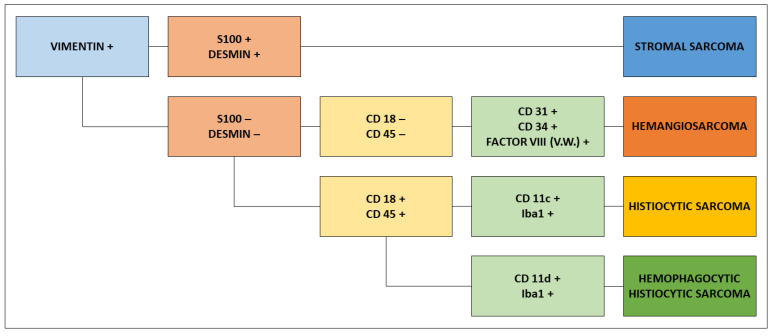
Differential diagnosis algorithm for splenic sarcomas based on immunohistochemical markers.

**Table 1 cancers-15-02025-t001:** Proposed clinical staging model for canine cutaneous HSA.

Primary tumor (T)	
T0	No evidence of tumor
T1	Primary tumor confined to the dermis
T2	Primary tumor involving the hypodermis, with or without concomitant dermal involvement. No muscle involvement
T3	Any primary tumor with muscle involvement
Regional Lymph Nodes (N)	
N0	No involvement of regional lymph nodes
N1	Involvement of regional lymph nodes
N2	Distant lymph node involvement
Distant Metastasis (M)	
M0	No evidence of distant metastases
M1	Presence of distant metastases
Staging	
I	T0 or T1; N0; M0
II	T2; N0, N1 or N2; M0
III	T1, T2 or T3; N0, N1 or N2; M1

Adapted from Mullin e Clifford (2020) [1] and Ward et al. (1994) [3].

**Table 2 cancers-15-02025-t002:** Chemotherapy protocols evaluated in canine subcutaneous and intramuscular HSA, type of surgery, and median survival.

Chemotherapy Protocol	N	HSA Type	Surgery	Mean Survival Time (Days)
VAC Protocol (Hammer et al., 1991) [32]	4	Subcutaneous	Incomplete resection (n = 4)	436
AC Protocol (Sorenmo et al., 1993) [23]	5	Subcutaneous	Complete resection (n = 2); Incomplete resection (n = 3)	240
Doxorrubicin +/− Cyclofosfamide (Bulakowski et al., 2008) [25]	21	Subcutaneous (n = 17) Intramuscular (n = 4)	Complete resection (in the first surgery—n = 11; in the second surgery—n = 5); Adjuvant radiotherapy after incomplete resection (n = 5) ^a^	1189 (subcutaneous) 272 (intramuscular) ^b^
Doxorrubicin +/− ciclofosfamide, vincristine or lomustine (Shiu et al., 2011) [26]	36	Subcutaneous (n = 55) Intramuscular (n = 16)	Complete resection (n = 18); Incomplete resection (n = 18)	212 (subcutaneous) 136 (intramuscular) ^c^

^a^ Adequate local control achieved for all dogs. ^b^ Presence of statistically significant difference between groups (*p* = 0.002). ^c^ Absence of statistically significant difference between groups (*p* = 0.268).

**Table 3 cancers-15-02025-t003:** Prognostic factors associated with non-visceral HSAs.

	Positive	Negative
Tissue of origin	Dermis (stage I)	Subcutaneous and muscle (stage II and III)
Subcutaneous or muscular invasion	Absent	Present
Actinic lesions	Present	Absent
Breeds	Predisposed breeds	Non-predisposed breeds
Surgical margins	Complete	Incomplete
Tumor size	<4 cm	>4 cm

**Table 4 cancers-15-02025-t004:** Proposed clinical staging model for canine visceral HSAHSA.

Primary Tumor (T)	
T0	No evidence of tumor
T1	Tumor less than 5 cm in diameter, confined to one organ
T2	Tumor greater than or equal to 5 cm in diameter, ruptured
T3	Tumor with invasion of adjacent structures
Regional Lymph Nodes (N)	
N0	No involvement of regional lymph nodes
N1	Involvement of regional lymph nodes
N2	Distant lymph node involvement
Distant Metastasis (M)	
M0	No evidence of distant metastases
M1	Presence of distant metastases
Staging	
I	T0 or T1; N0; M0
II	T2; N0, N1 or N2; M0
III	T1, T2 or T3; N0, N1 or N2; M1

Adapted from Mullin and Clifford (2020) [1].

**Table 5 cancers-15-02025-t005:** Simplified clinical staging model proposed for canine visceral HSA.

Primary Tumor (T)	
T0	No evidence of tumor
T1	Tumor confined to one organ, without evidence of rupture
T2	Tumor with evidence of rupture
Regional Lymph Nodes (N)	
N0	No involvement of regional lymph nodes
N1	Confirmed lymph node metastasis
Metástases distantes (M)	
M0	No evidence of distant metastases
M1	Presence of distant metastases
Staging	
I	T0 or T1; N0; M0
II	T1; N1; M0; or T2; N0; M0
III	T1, T2 or T3; N0, N1 or N2; M1

Adapted from Wendelburg et al. (2015) [33] and Bray et al. (2017) [143].

**Table 6 cancers-15-02025-t006:** Prognosis and mean survival time of dogs affected by splenic HSA, according to clinical staging.

Clinical Staging	Median Survival Time (Days)					
	Sorenmo et al. 1993 [23]	Kim et al. 2007 [150]	Wendelburg et al. 2015 [33]	Matsuyama et al. 2017 [151]	Batschinski et al. 2018 [146]	Ciepluch et al. 2018 [168]
I	250	345	160	259	196	193
II	186	93	60	125	117	105
III	87	68	30	62	23	63

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
