# Peer review of "Diagnosis, Prognosis, and Treatment of Canine Hemangiosarcoma: A Review Based on a Consensus Organized by the Brazilian Association of Veterinary Oncology, ABROVET"

_cancers, 2023, doi:10.3390/cancers15072025_

Round 1

Reviewer 1 Report

Accepted in present form for publication

Author Response

Dear reviewer, thank you so much for your time reviewing our manuscript. Thank you so much for considering our manuscript suitable for publication in the current form.

Reviewer 2 Report

no special comments

Author Response

(The authors gave the same response as above.)

Reviewer 3 Report

This paper (cancers-2255161) is a valuable article that summarizes the various views of veterinarians in Brazil on angiosarcoma, a refractory disease in dogs. This paper integrates information from empirical data and a huge of literature data to present extensive knowledge on the pathogenesis, epidemiology, diagnosis, and treatment of canine angiosarcoma. 

Minor revisions are requested below to further brush up on this paper.

Lines 169-175: It is not appropriate to mention tp53 mutations as a cause in canine tumors, since they occur frequently in all cancer types; other genetic mutations such as PIK3CA should also be mentioned.

Headings in Tables 4 and 5 must be in English

Correct the character sequence in Table 6 because it is out of alignment.

Author Response

Dear reviewer, thank you so much for your time reviewing our manuscript and we really appreciated you positive and constructive criticism for our manuscript improvement. Thank you for considering our paper for publication after minor revisions. Please, see the specific answer to the comments below:

Lines 169-175: It is not appropriate to mention tp53 mutations as a cause in canine tumors, since they occur frequently in all cancer types; other genetic mutations such as PIK3CA should also be mentioned.

Answer: Dear reviewer, we have included some additional information regarding TP53 and PI3K pathways as suggested.

Headings in Tables 4 and 5 must be in English

Answer: Dear reviewer, we are so sorry for this mistake. We have adjusted as suggested. All modification are highlighted in red.

Correct the character sequence in Table 6 because it is out of alignment.

Answer: Dear reviewer, we are so sorry for this mistake. We have adjusted as suggested. All modification are highlighted in red